# Primary Implant Stability Analysis of Different Dental Implant Connections and Designs—An In Vitro Comparative Study

**DOI:** 10.3390/ma15093072

**Published:** 2022-04-23

**Authors:** Perry Raz, Haya Meir, Shifra Levartovsky, Alon Sebaoun, Ilan Beitlitum

**Affiliations:** 1Department of Periodontology and Dental Implants, The Maurice and Gabriela Goldschleger School of Dental Medicine, The Sackler Faculty of Medicine, Tel Aviv University, Tel Aviv 6139001, Israel; hayameir@012.net.il (H.M.); alon.sebaoun@gmail.com (A.S.); beilan1612@gmail.com (I.B.); 2Department of Oral Rehabilitation, The Maurice and Gabriela Goldschleger School of Dental Medicine, The Sackler Faculty of Medicine, Tel Aviv University, Tel Aviv 6139001, Israel; shifralevartov@gmail.com

**Keywords:** primary stability, implant stability, ISQ, RFA, insertion torque, implant design, bone density

## Abstract

Primary implant stability can be evaluated at the time of placement by measuring the insertion torque (IT). However, another method to monitor implant stability over time is resonance frequency analysis (RFA). Our aim was to examine the effect of bone type, implant design, and implant length on implant primary stability as measured by IT and two RFA devices (Osstell and Penguin) in an in vitro model. Ninety-six implants were inserted by a surgical motor in an artificial bone material, resembling soft and dense bone. Two different implant designs—conical connection (CC) and internal hex (IH), with lengths of 13 and 8 mm, were compared. The results indicate that the primary stability as measured by RFA and IT is significantly increased by the quality of bone (dense bone), and implant length and design, where the influence of dense bone is similar to that of CC design. Both the Osstell and Penguin devices recorded higher primary implant stability for long implants in dense bone, favoring the CC over the IH implant design. The CC implant design may compensate for the low stability expected in soft bone, and dense bone may compensate for short implant length if required by the anatomical bone conditions.

## 1. Introduction

Osseointegration is an accepted histological term that refers to direct bone apposition over the implant surface under functional load. The clinical assessment of osseointegration is predominantly mechanical and depends on the primary and secondary stability of the implant [1,2,3]. Primary stability is defined as the absence of implant mobility at the bone site during implant insertion, while secondary stability requires bone formation and remodeling at the implant–bone interface. Secondary stability is positively correlated with primary stability [4,5]. 

The primary stability of an implant is influenced by the quality and quantity of bone, the drilling protocols, and the surgical modifications designed to compress the bone during implant placement, as well as by the macro topography of the implant [6,7,8,9,10,11].

Primary stability, which is a critical prerequisite for immediate loading, may be measured at implant placement by the insertion torque tool (IT, N/cm). This is a common and simple technique for which there is no need for additional instrumentation since the clinician uses this method routinely. Implant stability can then be subsequently evaluated at different time points during the healing process by additional non-invasive tools before implant rehabilitation [12,13].

A widely used method for monitoring implant stability during healing is based on resonance frequency (RFA) and essentially involves a bending test applied to the implant–bone interface. Devices using this method contain a transducer peg, which is connected to the implant and excited by magnetic waves over a range of frequencies. The frequency of the resultant vibration is automatically translated into an index called the implant stability quotient (ISQ), with values ranging between 0 and 100. The RFA values are a measure of the deflection of the implant–bone complex by the lateral forces applied by the transducer and reflect the multidirectional fixation strength [14]. This is a noninvasive and repeatable technique, which can be performed longitudinally to provide information about the stability during the healing process [15]. The ISQ value has been reported to correlate with the percentage of bone tissue in intimate contact with the implant surface, since primary/mechanical stability leads to more efficient achievement of secondary/biological stability [16]. Two common RFA tools are the Osstell (Osstell, Integration Diagnostics B, Göteborg, Sweden) and the Penguin (Integration Diagnostics Sweden AB, Gothenburg, Sweden) devices. The main difference between them is the excitation wave used, where the Penguin is considered to be an electronic device, while the Osstell system is a magnetic detection device [17,18]. RFA values are influenced by various factors: the design of the transducer itself; the stiffness of the implant fixture and its interface with the tissues and surrounding bone; and the total effective length above the marginal bone level [19,20].

Our previous work on the same in vitro model concluded that there is a strong positive correlation between IT and ISQ as measured by both the Osstell and Penguin devices. This correlation was not affected by bone type or implant length but was affected by implant design [21].

Our hypothesis is that longer implants in dense bone have a higher primary implant stability than short implants in soft bone regardless of implant design, namely, conical connection (CC) vs. internal hex (IH). It is possible that combinations of other parameters may have an additive positive influence on primary implant stability. 

The novelty of our research is that although we employ an in vitro model, it enables us to isolate the effect of individual variables (bone type, implant length, implant design) and also allows us to assess the additive effect of combinations of these variables on primary implant stability.

**The objective of our study** is to examine the individual and combined effect of bone type, implant length, and implant design on implant primary stability as measured by IT and ISQ measurements with two devices in an in vitro model.

## 2. Materials and Methods

This model is identical to the in vitro model we described previously [21]. Briefly, 96 implants (MIS, implants technologies) were inserted in an artificial bone material made of synthetic polyurethane 120 × 170 × 42 mm^3^ foam blocks (Sawbones, Malmö, Sweden) with different cortical thicknesses and trabecular densities [8]. The density of the soft bone blocks (#10) is 0.16 gcm^−3^ and they are laminated on one side with 1.5 mm dense cortical bone (# 50), which has a density of 0.8 g/cm^−3^. The density of the dense bone blocks (#40) is 0.64 g cm^−3^ and they are laminated on one side with 2 mm of cortical bone (#50). As a comparison, the mineral bone density of the posterior maxilla is 0.31 gcm^−3^, and that of the anterior maxilla is 0.55 gcm^−3^. The cortical thickness of the mandible is 2.22 mm, while the thickness in the maxilla is 1.49 mm [22]. The choice of a slightly softer and slightly denser block density was made in order to examine extreme cases. Two implant designs were compared. The first (IH) is a tapered internal hex (Seven^®^ new design MIS^®^ Implants Technology Ltd. Or Yehuda, Israel), and the second (CC) is less tapered with a conical connection (C1^®^ MIS^®^ Implants Technology Ltd. Or Yehuda, Israel). IH is characterized by a tapered design with an inter-thread distance of 2 mm, while CC is less tapered with an inter-thread distance of 1.5 mm, thereby providing more threads in this conformation. Additional differences are that CC has two spiral channels in its apex compared to three channels in IH, CC has a conical connection, while IH has an internal hexagon connection, and that the threads in IH are deeper than in CC. Common design features include threads that condense at the neck and cut at the apex, a domed apex, and a platform switching micro-gap. The implants were inserted to two different lengths of 13 or 8 mm at a constant distance of 30 mm from each other (Figure 1). There were 8 experiment groups, with 12 implants in each group. 

Primary stability measurements: The IT (N/cm) at implant insertion was evaluated by using a surgical motor with torque control (Implanted, W&H, Burnoose, Austria). The peak insertion torque was recorded from the surgical device display. The stability of each implant (one measurement from each of the 3 different directions) was measured with the Osstell (Integration Diagnostics AB, Göteborg, Sweden) and Penguin (Penguin Integration Diagnostics, Göteborg, Sweden) RFA devices after the transducer (smartpeg) was screwed to the implant to obtain the ISQ. The triplicate measurements from three different directions for each RFA device were performed after the implant was fully inserted manually with a ratchet device and was flush with the bone block (defined as no. 2). 

## 3. Statistical Analysis

Comparisons between the effect of the tested parameters were analyzed by a linear regression model in order to assess the interactions between the different parameters (bone, length, design) with each device. The multiple comparison correction of Benjamini and Hochberg was also applied. All analyses used “R” software version 4.0.5., and differences were accepted as being significant at *p* value < 0.05. *p* < 0.01 was also noted.

## 4. Results

The results reveal a significant effect of bone type, implant length, and implant design on the primary stability. Dense bone, an implant length of 13 mm, and CC implant design all significantly increased the RFA primary stability as measured by RFA and IT devices. The effect of dense bone was similar to that of CC implant design, while the slightly higher IT measurements for 8 mm long implants were attributed to the observation that the 13 mm implants were not fully inserted when measured. The effects of bone density and implant design on ISQ average were higher than that of implant length (Table 1).

The differences in the primary stability as measured by the different devices comparing the combined effect of the variable parameters (bone type, implant length and design) are presented in Figure 2.

All the comparisons were significantly different, with the exception of the ISQ measurements for IH in dense bone versus CC in soft bone. The biggest difference is exhibited by the RFA measurements of the IH and CC implants in soft bone as measured by the Penguin and Osstell devices (19.87 and 17.23 ISQ units, respectively). This reflects an additive effect of the CC design and the effect of dense bone. There were minor and non-significant differences between the stability of the IH implant in dense bone and a CC implant in soft bone. (Figure 3, Table 2).

A comparison of measurements by the different devices revealed that primary stability is higher by 9.6–11.1 ISQ units in dense as compared to soft bone. There was also an effect of implant length with a difference of 3.6 and 5.1 ISQ units in dense and soft bone, respectively, when comparing implants with a length of 13 or 8 mm. The parameters or implant length and bone type are additive and provided a maximum difference of 14.8 ISQ units in favor of a long implant in dense bone versus a short implant in soft bone. In dense bone, the implant length has a significant but low effect on the primary stability as measured by all the used devices (IT, O2, and P2, Figure 4, Table 3).

When analyzing the combined effect of implant design and length on the primary stability as measured by the different devices, the biggest difference in stability was in favor of long CC versus short IH implants. The results indicate that although implant length influences primary stability, the implant design influence is stronger in short implants (10.22) as opposed to long implants (8.71), as can be seen in the ISQ measurements by the Penguin device (Figure 5, Table 4). 

## 5. Discussion

The aim of this research was to examine the primary stability of implants in different mechanical settings and to evaluate the contribution of various parameters (bone type, implant length, and implant design), both individually and in combination, to the primary stability. This was achieved by IT and ISQ measurements made with two devices in an in vitro model. Our main findings are that each of the parameters—dense bone, an implant length of 13 mm, and CC design—increases the RFA primary stability significantly as measured by all the devices. The effect of dense bone is similar to the effect of CC implant design. The results reveal an additive effect of the CC implant design on the dense bone effect. The effects of bone type and implant length are also additive, with a long implant in dense bone being more stable than a short implant in soft bone. The combined effect of implant length and type on the primary stability as measured by the different devices exhibits the highest difference in favor of a long CC versus a short IH implant.

In accordance with our results, Farronato et al. used a similar model of rigid polyurethane foam blocks to show that higher IT and RFA values were associated implant length (3.6 N/cm increase per mm in length) and substrate density. They also reported that the drilling protocol affects the primary stability but that implant diameter had no effect [23].

A study of the RFA primary stability of short and extra-short dental implants inserted in an ex vivo pig ribs model concluded that tapered design and larger implants had higher primary stability in terms of ISQ values [24]. Therefore, it could be clinically beneficial to prefer a long implant (within limits) and a conical wall design in a severely resorbed low-quality bone ridge.

Falco et al., who implanted 120 different implant designs in various animal bone densities concluded that implant geometries and bone density are the main factors involved in the degree of primary implant stability [25]. In accordance with these findings, we can attribute the favorable primary stability seen with the CC implant macro-geometry compared to the IH design to the higher number of threads and the milder tapering.

The results of our study reveal statistically significant correlations between bone density and insertion torque values. This agrees with the findings of a previous study on human cadaver mandibles, where 24 implants were inserted into anterior and posterior regions of the jaw, and the bone densities were preoperatively determined using computerized tomography (CT). The maximum insertion torque values were recorded, and RFA primary implant stability measurements were performed [26].

In another study, RFA values were slightly but significantly decreased during 15 weeks of healing of implants placed in one stage in dense mandibular bone [27]. This observation was attributed to marginal bone loss and an increase in implant length above the bone crest [27]. 

A review of the clinical literature on the Resonance Frequency Analysis (RFA) techniques suggested that factors such as bone density, upper or lower jaw, abutment length and supracrestal implant length seem to influence RFA measurements. However, the data suggest that high RFA values indicate successfully integrated implants and that low/decreasing RFA values may be signs of ongoing disintegration and/or marginal bone loss. The prognostic value of the RFA in predicting loss of implant stability has yet to be established in prospective clinical studies [28].

A correlation was observed between bone quality and ISQ values, and primary stability was affected by the jaw and the bone type. The ISQ was greater in the mandible than the maxilla. Furthermore, the implants were placed in the anterior part of the mouth, either in the maxilla or in the mandible, showing greater values than those placed in the posterior areas of the mouth. The ISQ was found to be significantly higher in type I bone than in type III bone. The implant position, implant length and implant diameter did not affect primary stability [29,30,31]. Other studies failed to demonstrate such a correlation [32]. One study that documented the influence of gender, implant diameter and implant location on RFA values reported decreasing ISQ values with increasing implant lengths. This negative correlation may be explained by the fact that long implants may have a reduced diameter at the coronal end [31].

More in vitro studies that used artificial bone blocks aimed to evaluate the effects of implant designs on primary stability in different bone densities and bony defects found that three-wall and one-wall defects usually did not provide significant loss of primary stability, but a significant loss of stability was found when implants were inserted into circular defects. Implants with a more aggressive thread design could increase primary stability [33]. In bovine bone, it was showed that both ISQ and IT increased as the preparation technique reduces the implant site diameter when compared with the standard preparation. Tapered implants always showed higher or the same ISQ and IT values when compared with the cylindrical implant [34]. Additional factors that were demonstrated to affect implant stability in ex vivo models are clinician experience and the preparation technique [35]. 

Our model is an in vitro model that allows us to conduct a robust mechanical assessment of the parameters affecting primary implant stability without any biological variance bias. This is a standardized model with constant conditions, independent of clinician or patient variability. The obvious disadvantage of this model is the lack of the biological impact of bone properties. In future studies, it would also be useful to examine the isolated effect of implant internal connection on primary stability measurements.

## 6. Conclusions and Clinical Relevance

Primary stability measurements made by Osstell and Penguin devices reveal higher values for longer (13 mm) compared to shorter implants (8 mm), for dense bone compared to soft bone, and for implants with a CC rather than an IH design.The increased stability of the CC implant design may compensate for the low primary implant stability expected in soft bone.The effect of implant length in dense bone is significant but smaller than in soft bone. Dense bone may compensate for 8 mm implant insertion.The CC implant design may be used to enhance the primary implant stability in cases where insertion of an 8 mm implant is obligatory.

## Figures and Tables

**Figure 1 materials-15-03072-f001:**
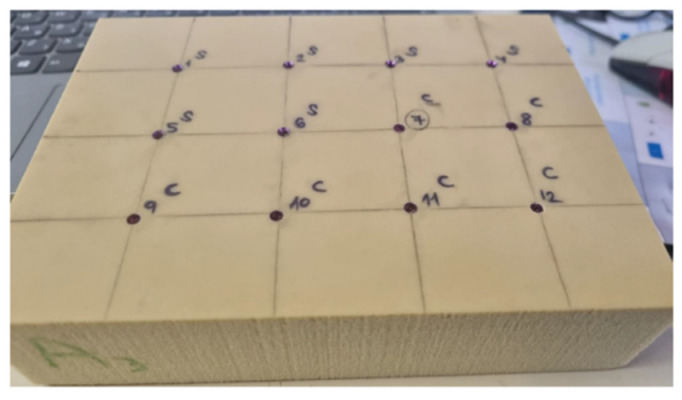
The experiment model of inserted implants in artificial bone block material made of synthetic polyurethane foam blocks.

**Figure 2 materials-15-03072-f002:**
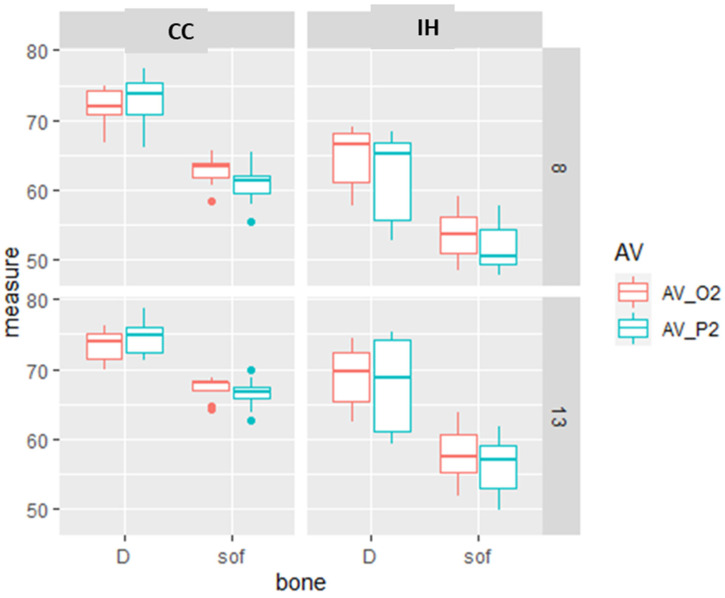
Box and whiskers plot of the RFA measurements of Osstell (O2) and Penguin (P2) in the D = dense and sof = soft bone type, in CC and IH implant design and 8 or 13 mm implant length. The box represents the interquartile range and the median, and the whiskers represent the SD. There are some values that are outliers.

**Figure 3 materials-15-03072-f003:**
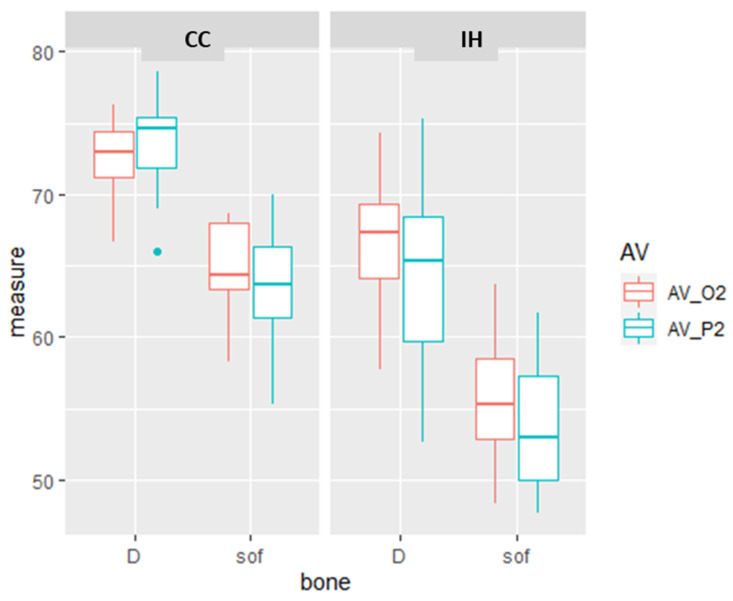
The effect of implant design and bone type on implant stability. Box and whiskers plot of the RFA measurements of Osstell (O2) and Penguin (P2) in the D = dense and sof = soft bone type, in CC and IH implant design. The box represents the interquartile range and the median, and the whiskers represent the SD. There are some values that are outliers.

**Figure 4 materials-15-03072-f004:**
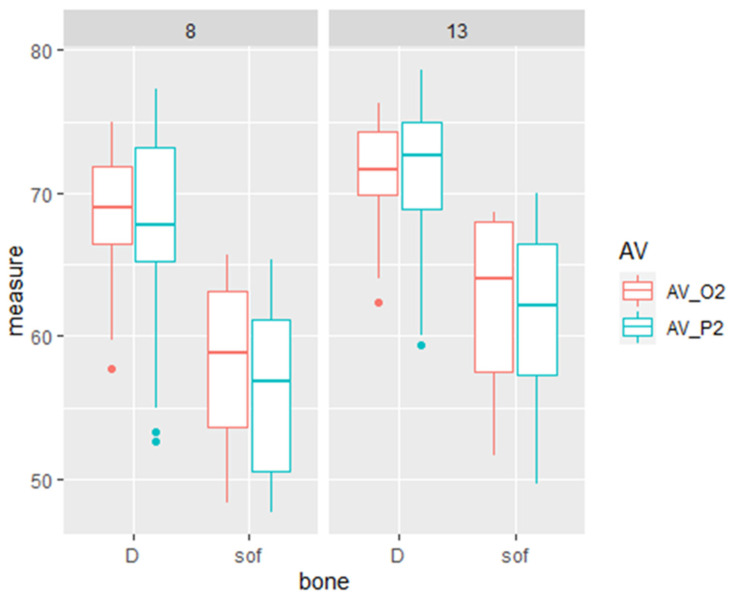
The effect of implant length and bone type on implant stability. Box and whiskers plot of the RFA measurements of Osstell (O2) and Penguin (P2) in the D = dense and sof = soft bone type, in different implant length 8 or 13 mm. The box represents the interquartile range and the median, and the whiskers represent the SD. There are some values that are outliers.

**Figure 5 materials-15-03072-f005:**
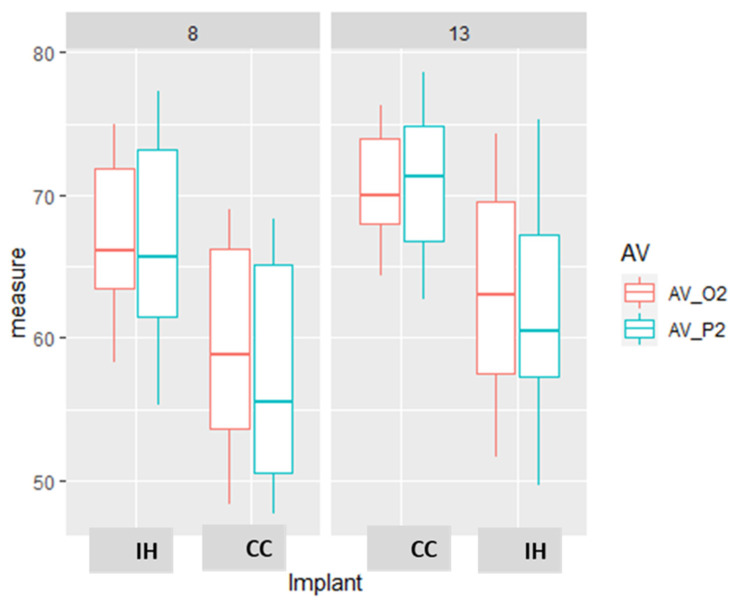
Box and whiskers plot of RFA measurements of Osstell (O2) and Penguin (P2) in CC or IH implant design, with different implant length (8 or 13 mm). The box represents the interquartile range and the median, and the whiskers represent the SD.

**Table 1 materials-15-03072-t001:** ANOVA of the difference of average of implant stability measurements between the tested parameters of bone type (S = soft, D = dense) or implant length (13 or 8 mm) or implant design (CC = conical connection, IH = internal hexagon) by the IT, Osstell (O2), and Penguin (P2) devices. *p* < 0.05 *; *p* < 0.01 **.

Difference	IT (N/Cm)	AV_O2 (ISQ)	AV_P2 (ISQ)
**Bone (D-S)**	2.566 **	9.669 **	10.47 **
**Length (13–8)**	−1.619 *	3.521 **	4.402 **
**Implant (CC-IH)**	6.729 **	7.607 **	9.491 **

**Table 2 materials-15-03072-t002:** The difference of average implant stability measurements between the tested parameters of bone type (soft or dense) and implant type (IH or CC) by the IT, Osstell (O2), and Penguin (P2) devices.

Diff	*p* < 0.05 * *p* < 0.01 **	IT (N/Cm)	AV_O2 (ISQ)	AV_P2 (ISQ)
Bone& Implant design	**sof:CC-D:CC**	−4.111 **	−7.836 *	−10.05 *
	**D:IH-D:CC**	−8.346 **	−5.856 *	−9.173 *
	**sof:IH-D:CC**	−9.189 **	−17.23 *	−19.87 *
	**D:IH-sof:CC**	−4.235 **	1.98	0.8741
	**sof:IH-sof:CC**	−5.078 **	−9.398 *	−9.818 *
	**sof:IH-D:IH**	−0.8429	−11.38 *	−10.69 *

**Table 3 materials-15-03072-t003:** The difference of average implant stability measurements between the combined tested parameters of bone type and implant length as measured by the IT, Osstell (O2), and Penguin (P2) devices.

Diff	*p* < 0.05 * *p* < 0.01 **	IT (N/Cm)	AV_O2 (ISQ)	AV_P2 (ISQ)
Bone & length	**sof:8–D:8**	−3.362 *	−10.45 *	−11.14 *
	**D:13–D:8**	−2.415	2.679	3.665 *
	**sof:13–D:8**	−4.167 **	−6.071 *	−5.983 *
	**D:13–sof:8**	0.9477	13.13 *	14.81 *
	**sof:13–sof:8**	−0.8048	4.384 *	5.158 *
	**sof:13–D:13**	−1.753	−8.75 *	−9.647 *

**Table 4 materials-15-03072-t004:** The difference of average implant stability measurements between the combined tested parameters of CC or IH implant design and implant length as measured by IT, Osstell (O2) and Penguin (P2) devices.

Diff	*p* < 0.05 * *p* < 0.01 **	IT (N/Cm)	AV_O2 (ISQ)	AV_P2 (ISQ)
Length & Implant design	**13:CC–8:CC**	0.3635	2.907 *	3.548 *
	**8:IH–8:CC**	−4.716 **	−8.126 *	−10.22 *
	**13:IH–8:CC**	−8.515 **	−4.148 *	−5.162 *
	**8:IH–13:CC**	−5.08 **	−11.03 *	−13.77 *
	**13:IH–13:CC**	−8.878 **	−7.054 *	−8.71 *
	**13:IH–8:IH**	3.798 **	3.979 *	5.063 *

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
