# Peer review of "Primary Implant Stability Analysis of Different Dental Implant Connections and Designs—An In Vitro Comparative Study"

_materials, 2022, doi:10.3390/ma15093072_

Round 1
Reviewer 1 Report
The manuscript entitled “The contribution of implant design to primary implant stability– a comparative in vitro study of conical connection Vs. internal hex implant designs” submitted to Materials aims to analyze the influence of many implant features on primary implant stability .
The manuscript appears interesting, nevertheless the text structure must be revised in a more orderly way, following the types of article according to the journal guidelines.
- English language: Maior spell check is required. The grammatical part needs to be revised well. The article is not written in fluent, scientific English.
- Abstract: this part needs a thorough revision. Reduce the introduction part and join the aim of the manuscript by describing it in better English. Results: the results obtained should be listed numerically, again the English should be revised. Conclusion: the English form should be revised and described in a better way.
- Article structure - references go between [ ] and added before the end of the sentence (before the full stop). References should be listed in the bibliography following the Mendeley method.
It seems that the authors have never written a scientific article.
The figures are incorrectly positioned and the text of the article seems redundant and not good enough to be evaluated.
Although the research seems very interesting I suggest the authors revise the text thoroughly, following the structure of a scientific article.
I also encourage the authors to carry out a thorough review of the English form before eventually resubmitting the manuscript.
Author Response
Please enclosed reply letter to the reviewer comments

Reviewer 2 Report
INTRODUCTION.
1- THE CONCEPT OF OSSEOINTEGRATION, AS INITIALLY DESCRIBED BY BRANEMARK, REQUIRES BONE/IMPLANT CONTACT UNDER LOAD
MATERIAL AND METHODS
1- THE AUTHORS SHOULD PROPERLY CHARACTERIZE THE IMPLANTS USED. TERMS LIKE MORE OR LESS TAPERED ARE NOT PRECISE.
2- I SUGGEST CHANGE THE SPECIFIC TERMS FOR THE GROUPS. IH AND CC DO NOT REPRESENT ALL THE DIFFERENCES BETWEEN THE IMPLANTS STUDIED
CONCLUSION
1-ALTHOUGH THERE ARE DIFFERENT DEFINITIONS OF SHORT IMPLANTS IN THE LITERATURE, THERE IS NO CONSENSUS THAT 8 MM IMPLANTS CAN BE CONSIDERED SHORT, THEREFORE I SUGGEST TO USE THE EXACT IMPLANT LENGHT IN THE CONCLUSION OF THE STUDY INSTEAD OF SHORT IMPLANT
Author Response
Please enclosed a file with response to the reviewer comments

Round 2
Reviewer 1 Report
Authors improved manuscript form.
I suggest other changes to the text to make the article scientifically assessable:
Title:
I suggest to change the title into "An in vitro comparative study on synthetic models of different dental implant connections and designs: analysis of primary implant stability"
Discussion:
As this is an in vitro study, I recommend referring mainly to in vitro studies when comparing with the literature.
I recommend supplementing the text with recent references on in vitro primary stability analysis [PMID: 34639933 - PMID: 32475099 - PMID: 32575702 - doi.org/10.3390/app10238623 - PMID: 33007841].
I recommend inserting references following the correct Mendeley format.
After the suggested corrections, I expressly advise authors to resubmit the text of the manuscript by deleting the previously deleted parts so that the text can be assessed more clearly.
